# Recent Developments in Direct C–H Functionalization of Quinoxalin-2(1*H*)-Ones via Multi-Component Tandem Reactions

**DOI:** 10.3390/molecules28062513

**Published:** 2023-03-09

**Authors:** Qiming Yang, Biao Wang, Mian Wu, Yi-Zhu Lei

**Affiliations:** 1Guizhou Provincial Key Laboratory of Coal Clean Utilization, School of Chemistry and Materials Engineering, Liupanshui Normal University, Liupanshui 553004, China; 2Henan Key Laboratory of New Optoelectronic Functional Materials, College of Chemistry and Chemical Engineering, Anyang Normal University, Anyang 455000, China

**Keywords:** multi-component reaction (MCR), quinoxalin-2(1*H*)-ones, C–H functionalization, heterocyclic compounds

## Abstract

The direct C–H multifunctionalization of quinoxalin-2(1*H*)-ones via multicomponent reactions has attracted considerable interest due to their diverse biological activities and chemical profile. This review will focus on recent achievements. It mainly covers reaction methods for the simultaneous introduction of C–C bonds and C–R_F_/C/O/N/Cl/S/D bonds into quinoxalin-2(1*H*)-ones and their reaction mechanisms. Meanwhile, future developments of multi-component reactions of quinoxalin-2(1*H*)-ones are envisaged, such as the simultaneous construction of C–C and C–B/SI/P/F/I/SE bonds through multi-component reactions; the construction of fused ring and macrocyclic compounds; asymmetric synthesis; green chemistry; bionic structures and other fields. The aim is to enrich the methods for the reaction of quinoxalin-2(1*H*)-ones at the C3 position, which have rich applications in materials chemistry and pharmaceutical pharmacology.

## 1. Introduction

In multi-component reactions (MCRs), three or more substrate molecules are transformed in a single pot to produce the desired product. In some cases, they are also referred to as multi-component assembly processes (MCAPs). Through MCRs, a large number of organic compounds can be synthesized in an efficient and atom-economic manner, with little waste and no hazardous substances, and with high yields of final products [1,2,3]. MCRs were first proposed by Strecker in 1850 for the synthesis of α-amino nitriles from ammonia, carbonyl compounds and hydrogen cyanide [4,5,6,7]. Subsequently, more and more researchers have become interested in this approach, as this multiple bonding route involving C–C and C–heteroatom bond formation can be used to prepare a variety of heterocyclic compounds with combinatorial, medicinal and agricultural chemistry applications [8,9]. In recent years, multicomponent reactions using quinoxalin-2(1*H*)-ones as a substrate have gained increasing interest (Figure 1). A range of ingenious and innovative methods have been developed to offer prosperous prospects for this highly applicable compound.

Quinoxalin-2(1*H*)-ones have been considered as a specific class of heterocyclic structural groups existing in a diverse range of biologically active natural products and pharmaceutical compounds [10,11,12]. Furthermore, the quinoxalin-2(1*H*)-one fragment is a promising electron-withdrawing substituent for the development of push-pull systems with promising photophysical properties for various applications in material science [13,14]. In particular, the C3-substituted quinoxalin-2(1*H*)-one derivatives show a broad range of biological activities (Figure 2) [15,16,17,18,19,20,21,22,23,24]. The direct C–H functionalization of quinoxalin-2(1*H*)-ones at the C3 position is the most cost-effective way to synthesize a wide range of quinoxalin-2(1*H*)-one derivatives containing valuable functional groups. In recent years, quinoxalin-2(1*H*)-one has been prominently studied [15,16,17,18,19,20,21,22,23,24]. Furthermore, the direct acid-catalyzed C–H bond functionalization of quinoxalin-2(1*H*)-one based on a nucleophilic aromatic substituted hydrogen approach (S_N_^H^-reaction) is of equal interest [25,26,27,28,29]. Several reviews summarize the C-3 functionalization of quinoxalin-2(1*H*)-one and were published between 2019 and 2021 [15,16,17,18,19,20,21,22,23,24]. However, there are no reviews discussing the topical issue of the C–H functionalization of quinoxalin-2(1*H*)-one via multicomponent reactions. Our group has also conducted some studies on the functionalization of quinoxalin-2(1*H*)-one at the C3 position [30,31,32]. Therefore, a review is necessary in order to better develop the functionalization of quinoxalin-2(1*H*)-one at the C3 position via a multi-component tandem approach. This review focuses on the C–H functionalization of quinoxalin-2(1*H*)-one by multi-component tandem reaction methods in recent years and demonstrates the corresponding reaction mechanisms. We classify them according to the different chemical bonds formed and the simultaneous construction of both C–C and C–R_F_/C/O/N/Cl/S/D bonds via multi-component tandem reactions.

## 2. Direct C–H Functionalization of Quinoxalin-2(1*H*)-Ones via Multi-Component Tandem Reactions

### 2.1. Simultaneous Construction of Both C–C and C–R_F_ Bonds via Multi-Component Tandem Reactions

Organic fluorides are widely found in pharmaceuticals, agrochemistry, and materials science. The introduction of fluorine atoms into the organic framework not only produces profound changes in the physical and chemical properties of the target molecules, but also has special biological properties [33]. During the direct C–H bifunctionalization of quinoxalin-2(1*H*)-ones through multi-component tandem reactions, it is possible to construct bonds C–C bonds and C–R_F_ bonds in a single step. Recently reported reactions include trifluoroalkylation, difluoroalkylation and perfluoroalkylation.

#### 2.1.1. Trifluoroalkylation

In 2020, Wei et al. [34] reported a simple and efficient three-component protocol for the construction of various 3-trifluoroalkylquinoxalin-2(1*H*)-ones via K_2_S_2_O_8_-mediated difunctionalization of unactivated alkenes with quinoxalin-2(1*H*)-ones and CF_3_SO_2_Na under metal-free conditions (Figure 3). As soon as water was added to this reaction system, the reaction efficiency was evidently improved. The optimal reaction conditions were determined after screening a variety of oxidants and solvents. K_2_S_2_O_8_ proved to be the preferred oxidant, and a 4:1 ratio of CH_3_CN/H_2_O produced the highest quality product with 77% yield. Differently substituted quinoxalin-2(1*H*)-ones and unactivated alkenes were used to investigate substrate scope after the reaction conditions were established. Following a series of control experiments, the mechanism of the reaction was established. The addition of TEMPO resulted in the complete quenching of the reaction progress, confirming the presence of a radical mechanism. Furthermore, the radical addition product was detected by LC-MS when the radical capture reaction of ethene-1,1-diyldibenzene was conducted under standard conditions. As shown in Figure 3, a possible reaction mechanism has been proposed. At the beginning, K_2_S_2_O_8_ was used to produce CF_3_ radical **1e**. In subsequent steps, alkene **1b** was selectively added to the CF_3_ radical to generate alkyl radical **1f**. In turn, alkyl radical **1f** added to quinoxalin-2(1*H*)-one **1a** produced nitrogen radical **1g**. Then, intermediate **1g** was further oxidized by K_2_S_2_O_8_ to give a nitrogen cation intermediate **1h**. Finally, the desired product **1d** was obtained by the deprotonation of intermediate **1h**.

In 2020, Li et al. [35] also used CF_3_SO_2_Na, unactivated alkenes and quinoxalin-2(1*H*)-ones to complete a three-component reaction (Figure 4). The difference was the use of PhI(OAc)_2_ as the oxidant and ethyl acetate as the solvent. A variety of different unactivated alkenes and quinoxalin-2(1*H*)-ones are suitable for this reaction method. Maximum yields of 80% were achieved. The reaction mechanism was also found to be a free radical mechanism after controlled experiments, which is basically consistent with the reaction mechanism reported by Wei’s group (Figure 4).

In the same year, Huang et al. [36] reported an unexpected three-component reaction of quinoxalin-2(1*H*)-one, tert-butyl peroxybenzoate (TBPB) and hexafluoroisopropanol (HFIP) (Figure 5). A variety of hydroxyhexafluoroisobutylated quinoxalin-2(1*H*)-ones were formed under CuBr-catalyzed and TBPB-oxidized conditions. In screening the reaction conditions, the addition of K_2_CO_3_ resulted in a significant increase in yield. The range of substrates was extended with a maximum isolated yield of 72%. The methylene in the product was deduced from TBPB, and the reaction mechanism was proposed after controlled experiments and literature reports. TBPB was reduced by Cu(I) to produce the tert-butoxy radical. β-Scission of the tert-butoxy radical yielded acetone and a methyl radical, which was then added to quinoxalin-2(1*H*)-one **3a** to form the radical intermediate **3e**. On the other hand, a two-step hydrogen atom transfer (HAT) from HFIP to methyl generated hexafluoroacetone. Finally, the nucleophilic addition of **3f** and hexafluoroacetone produced product **3d** (Figure 5).

In the year 2021, Wei’s group [37] reported different photocatalytic methods for the same reaction (Figure 6). They have developed a facile and efficient visible-light-induced three-component reaction of quinoxalin-2(1*H*)-ones, alkenes and CF_3_SO_2_Na, leading to various 3-trifluoroalkylated quinoxalin-2(1*H*)-ones at room temperature. The present photocatalytic tandem reaction, which utilizes 4CzIPN (1,2,3,5-tetrakis(carbazol-9-yl)-4,6-dicyanobenzene) as the photocatalyst and air as the green oxidant, offers a mild and environmentally friendly protocol to access a number of 3-trifluoroalkylated quinoxalin-2(1*H*)-ones in moderate to favorable yields. Through this methodology, various 3-trifluoroalkylated quinoxalin-2(1*H*)-ones could be obtained in moderate to high yields from simple and readily available materials in the absence of any strong external oxidant. Following a series of control experiments, the mechanism of the reaction was established. Moreover, Stern–Volmer quenching experiments were conducted to prove an energy transfer process between 4CzIPN and substrates under visible-light irradiation. As shown in Figure 6, a possible reaction pathway was proposed. In the first step, 4CzIPN* was excited from 4CzIPN by visible-light irradiation. Afterwards, 4CzIPN^•−^ and radical cation **4e** were produced by a single electron transfer (SET) from **4a** to the excited state of 4CzIPN*. 4CzIPN^•−^ was oxidized by dioxygen to give the ground state 4CzIPN and O_2_^•−^. At the same time, CF_3_SO_2_Na **4c** was oxidized by O_2_ (air) to produce the O-centered radical **4f**. Radical **4f** underwent self-induced loss reduction in SO_2_ to give CF_3_ radical **4g**. Subsequently, **4g** was added to the olefin **4b** to form the alkyl radical **4h**, which attacked the radical cation **4e** and formed a nitrogen cation **4i**. Lastly, the deprotonation of **4i** produced the desired product **4d**.

In the same year, the Huang team [38] reported a three-component tandem reaction for the photocatalytic bifunctionalization of olefins using quinoxalin-2(1*H*)-ones and ICH_2_CF_3_/ICH_2_CF_2_H as feedstocks and Ir(ppy)_2_(dtbbpy)PF_6_ (ppy = 2-phenylpyridine, dtbbpy = 4,4′-di-tert-butyl-2,2′-bipyridine) as catalyst. (Figure 7). Various trifluoroacetylated and difluoroacetylated quinoxalin-2(1*H*)-ones were synthesized. In the substrate extension, the highest yield of the three-component reaction of trifluoroethylated quinoxalin-2(1*H*)-ones reached 77%, and the highest yield of the three-component reaction of difluoroethyl quinoxalin-2(1*H*)-ones reached 74%. After controlled experiments and literature analysis, they suggested that the mechanism of this three-component reaction might be a free radical cascade reaction, but no reaction details were given.

Graphene oxide (GO) has an abundance of defects and a wide range of oxygen functional groups that can be used in organocatalytic reactions. In 2021, Liu et al. [39] proposed a three-component tandem reaction using GO as a catalyst and CF_3_SO_2_Na as a powerful CF_3_ radical source to form olefinic C–CF_3_ with quinoxalin-2(1*H*)-ones and alkynes, resulting in a series of E/Z isomeric mixed 3-trifluoroalkyl quinoxalin-2(1*H*)-ones. Based on mechanistic studies, GO is capable of auto-tandem radical addition to trifluoromethyl and superoxide radicals (Figure 8).

In 2022, Shen et al. [40] developed a three-component reaction consisting of indole, quinoxalin-2(1*H*)-one and CF_3_SO_2_Na, using CuF_2_ as a catalyst and K_2_S_2_O_8_ as an oxidant, to easily obtain various 3-[2-(trifluoromethyl)1*H*-indol-3-yl]quinoxalin-2(1*H*)-ones (Figure 9). The strategy exhibited high site selectivity, step economy and a wide range of substrates. A possible reaction mechanism was suggested in Figure 9. A weak coordination between Cu(II)Ln species and the nitrogen atom of quinoxalin-2(1*H*)-one led to the generation of **7e**. By oxidative dehydrogenation, **7e** reacted with **7b** to form **7f**, which was then converted to **7h**. In addition, Cu(I)Ln was oxidized by K_2_S_2_O_8_ to Cu(II)Ln. By weak coordination of Cu(II)Ln species with the carbonyl group of the quinoxalin-2(1*H*)-one motif, **7h** combined with Cu(II)Ln to produce complex **7i**. As a result of the oxidation of CF_3_SO_2_Na and K_2_S_2_O_8_, a trifluoromethyl radical was generated. Then, the indole ring of complex **7i** was attacked by a CF_3_· to produce **7j**, which was transformed into intermediate **7k** through the single electron transfer (SET) process. Thereafter, the oxidation of **7k** occurred, leading to compound **7l** by oxidation. Eventually, the target product **7d** was obtained by the deprotonation of **7l** with the regeneration of Cu(II)Ln.

#### 2.1.2. Perfluoroalkyl

In 2019, Studer et al. [41] described the reaction of α-perfluoroalkyl-β-heteroaryl groups with perfluoroalkyl iodides and quinoxalin-2(1*H*)-ones on various alkenes initiated under visible light (Figure 10). This three-component free radical cascade reaction allowed the efficient synthesis of a series of quinoxalin-2(1*H*)-one derivatives containing perfluoroalkyl groups in moderate to excellent yields under mild conditions. The reactions proceed through acidic amino radicals that are readily deprotonated to give the corresponding radical anions capable of maintaining the radical chains. The radical chains act as reducing reagents for single electron transfer (Figure 10).

#### 2.1.3. Difluoroalkylation

In 2022, Lu et al. [42] published a photocatalytic three-component reaction consisting of quinoxalin-2(1*H*)-ones, alkenes and hypervalent iodine(III) reagents (Figure 11). They used an original difluoroiodine(III) reagent of their own design, showing high reactivity with [bis(trifluoroacetoxy)iodo]benzene (PIFA) as the oxidizing agent. Under mild conditions, a wide range of difluoroalkyl-substituted quinoxalin-2(1*H*)-ones can be obtained in up to 95% yield. The experimental mechanism suggests the involvement of a difluoroalkyl radical intermediate in this reaction (Figure 11).

In the same year, Li et al. [43] reported visible-light photoredox-catalyzed, unactivated olefin, quinoxalin-2(1*H*)-one and bromodifluorinated reagents involved in a three-component reaction via fac-Ir(ppy)_3_ catalysis (Figure 12). The difluoroalkyl-containing quinoxalin-2(1*H*)-ones were synthesized by the single-step formation of two novel C–C bonds under mild reaction conditions. Three-component reactions showed high functional group compatibility and a wide range of substrates. Free radical processes may be responsible for this three-component transformation, according to preliminary mechanistic studies (Figure 12).

### 2.2. Simultaneous Construction of Double C–C Bonds via Multi-Component Tandem Reactions

The establishment of C–C bonds by direct C–H functional grouping is more atomically economical than the establishment of C–C bonds based on transition metal-catalyzed cross-coupling reactions of halogenated hydrocarbons [44].

#### 2.2.1. Alkylation

In 2021, Yao et al. [45] described a metal-free catalytic, H_2_O_2_-mediated three-component reaction of quinoxalin-2(1*H*)-ones, DMSO and styrene. A series of 3-substituted quinoxalin-2(1*H*)-ones were synthesized using DMSO as the methylation reagent (Figure 13). The reaction substrates were well tolerated and the products were obtained in moderate to acceptable yields, with the highest yields up to 86%. The reaction was shown to be a free radical reaction mechanism by controlled experiments. Initially, hydroxyl radicals were generated through the homolytic cleavage of H_2_O_2_ and subsequently reacted with DMSO to produce methyl radicals **11e**. The methyl radical then selectively attacked styrene **11b**, generating alkyl radical **11f**. As a result of the addition of alkyl radical **11f** to quinoxalin-2(1*H*)-one **11a**, a nitrogen radical **11g** was formed, which was then oxidized by H_2_O_2_, resulting in a nitrogen cation intermediate **11h**. In the end, intermediate **11h** was deprotonated to produce the desired product (Figure 13).

In 2022, Baishya et al. [46] reported a three-component reaction of quinoxalin-2(1*H*)-ones with 4-hydroxycoumarin/4-hydroxy-6-methyl-2-pyrone and aryl ethylene (Figure 14). The 3-alkylquinoxalin-2(1*H*)-one was synthesized using K_2_S_2_O_8_ as the oxidant and DMSO as the solvent. During the optimization of the reaction conditions, the highest yield of the three-component product was only 44%. They performed substrate extensions with yields below 48%. The involvement of free radicals in the reaction process was demonstrated by controlled experiments. The reaction mechanism was essentially similar to the above reaction, with K_2_S_2_O_8_ initiating free radicals (Figure 14).

#### 2.2.2. Cyanoalkylation

In the same year, Baishya and his group [47] published a K_2_S_2_O_8_-mediated three-component reaction (Figure 15). The reaction of quinoxalin-2(1*H*)-ones with azobis(alkylcarbonitrile)s and aryl ethylenes was used to synthesize 3-cyanoalkyl quinoxalin-2(1*H*)-ones. This scheme introduced cyano into quinoxalin-2(1*H*)-ones and is widely used in reasonable yields and acceptable functional group tolerances. The replacement of aryl ethylene with phenylacetylene is also applicable to this method. Controlling experimental and literature knowledge, they proposed a radical reaction mechanism similar to the above alkylation (Figure 15). The reaction started by heating AIBN to form the 1,1-dimethyl-1-cyanomethyl radical **13e**, which added to the terminal position of the olefin to give the secondary alkyl radical **13f**. Then, **13f** attacked the C3 position of the quinoxalin-2(1*H*)-one **13a** to give the N-radical **13g** which, after H-atom abstraction by the excess SO_4_^•−^ present in the reaction medium, gave the three-component desired product **13d** (pathway a). Again, **13g** was deprotonated to form the radical anion **13h**. Then, **13h** reacted with persulphate via SET to give the desired product **13d** (pathway b).

### 2.3. Simultaneous Construction of Both C–C and C–O Bonds via Multi-Component Tandem Reactions

Ether functional groups constructed through C–C and C–O bonds are an essential structural characteristic of many natural products, pharmaceutical functional materials and agrochemicals [48].

In 2022, Yao et al. [49] implemented a simple strategy for the synthesis of 3-substituted quinoxalin-2(1*H*)-ones containing ether units via a three-component synthesis of quinoxalin-2(1*H*)-one, styrene, and tert-butyl peroxybenzoate (TBPB) using CuCl as a catalyst (Figure 16). This method introduced both C–C and C–O bonds in the product with acceptable functional group tolerance. After controlled experiments, a reaction mechanism was proposed in Figure 16. First, TBPB **14c** was homolytically cleaved under heating conditions to produce benzoyloxy radicals (BzO·) and tert-butoxy radicals **14e**. Simultaneously, Cu(I) was converted to Cu(II) and BzO^−^ was converted to benzoyloxy anion (BzO^−^) due to electron transfer. At the same time, **14e** selectively attacked styrene **14b**, leading to the generation of alkyl radical **14f**. Next, **14f** was added to **14a** to produce nitrogen radical **14g**. Then, **14g** provided the nitrogen cation **14h** via Cu(II) single-electron oxidation. Finally, the expected coupling product **14d** was provided by the deprotonation of **14h**.

In the same year, Wang et al. [50] reported the visible light-induced bifunctionalized three-component reaction of 2,3-dihydrofuran with quinoxalin-2(1*H*)-one and tert-butyl hydroperoxide (Figure 17). They synthesized a series of 2,3-disubstituted tetrahydrofurans in high yields by simultaneously forming C–C and C–O bonds using 2,4,6-triphenylpyrylium tetrafluorborate (TPPT) as a photocatalyst. The multicomponent reactions exhibited excellent site-selectivity of carbon–carbon double bonds, superior functional group tolerance and high diastereoselectivity (d.r. > 99%). The mechanism was determined by controlled experiments and hydroxyl radical assay as a visible light-induced radical reaction (Figure 17). First, the photocatalyst (PC^+^) was excited under visible light irradiation to generate the excited species (PC^+^*), which then underwent a single electron transfer (SET) process with substrate **15a** to afford a radical cation intermediate **15e** and a PC^•^. *tert*-Butyl hydroperoxide (TBHP) abstracted an electron from the obtained PC^·^ to give a HO^·^ radical, which could be detected by EPR, and a *^t^*BuO^−^ together with the oxidation of PC^•^ to PC^+^ for the catalytic cycle. On the other hand, A reacted with 2,3-dihydrofuran (**15b**) to give intermediate **15f**, which was attacked by TBHP as a nucleophilic reagent to give intermediate **15g**. In the presence of *^t^*BuO^−^, the deprotonation of **15g** gave the intermediate **15h**. Finally, **15h** underwent SET oxidation and deprotonation to give the desired product **15d**.

### 2.4. Simultaneous Construction of Both C–C and C–N Bonds via Multi-Component Tandem Reactions

#### 2.4.1. Azidoalkylation

It is highly attractive to synthesize organic azides by difunctionalizing alkenes, owing to the efficient and consecutive introduction of N_3_ and other functional groups [51]. Specifically, organic azides resulting from this process could undergo a variety of transformations and have a wide range of synthetic applications.

In 2019, Zhang et al. [51] reported for the first time a three-component tandem reaction of quinoxalin-2(1*H*)-ones with unactivated olefins and TMSN_3_ (Figure 18). The reaction used a hypervalent iodine(III) reagent as an oxidant to synthesize a series of quinoxalin-2(1*H*)-ones containing azide groups. Remarkably, this was an efficient transformation accomplished within 1 min. Through controlled experiments, a single electron transfer (SET) mechanism was revealed in Figure 18. Initially, TMSN_3_ was oxidized by PhI(TFA)_2_ to produce the azide radical **16e**. The resulting **16e** was then chemically selectively added to **16b** to yield an alkyl radical **16f**. Next, the addition of **16f** to substrate **16a** yielded the nitrogen radical intermediate **16g**. Thereafter, **16g** was converted to a carbon radical **16h** via a 1,2-hydrogen transfer process. Then, **16h** was further oxidized by PhI(TFA)_2_ to form a carbon cation intermediate **16i** via another single electron transfer (SET) process. Lastly, the desired product **16d** was provided by the deprotonation of **16i**.

In 2020, Qin et al. [52] published a cascade azidoalkylation reaction with a free radical mechanism (Figure 19). This three-component reaction used PhI(OAc)_2_ as an oxidant to provide a valuable series of β-azidoalkylated quinoxalinones by coupling olefins, quinoxalin-2(1*H*)-ones, and TMSN_3_. Notably, the reaction was completely blocked when the base was removed from the reaction system. Why the presence of bases was indispensable was not answered by the authors. There was a puzzling result in contrast to Zhang’s study [51] where the same oxidant (PhI(OAc)_2_), solvent (MeCN) and temperature (r.t.) gave 0% yield while Zhang’s result was 52% yield. The only difference was that Zhang was an open bottle reaction and Qin was N_2_ protected. A free radical reaction mechanism similar to that described above was proposed by controlled experiments and previous reports (Figure 19).

In 2022, the same Zhang group [53] demonstrated a visible light-mediated three-component tandem reaction method using 9-mesityl-10-methylacridinium perchlorate (Acr^+^MesClO_4_^−^) as a photocatalyst for the reaction of N-heterocyclic aromatic hydrocarbons with unactivated olefins and TMSN_3_ (Figure 20). This strategy achieved a room temperature conversion that was not achieved by previous authors and does not use a polyvalent iodine reagent. It is noteworthy that the reaction cannot be carried out under an N_2_ atmosphere. The method is very tolerant of functional groups and enables the synthesis of various heterocyclic olefins with medicinal value. The mechanism of the visible light-driven free machine reaction was proposed by controlled experiments. The overall process is consistent with the thermocatalytic reaction mechanism except that it starts with the photocatalytic initiation of the azide free machine and ends with O_2_^·−^ deprotonation to obtain the product (Figure 20).

#### 2.4.2. Oxime

In 2021, Zhang et al. [54] developed a multicomponent reaction for the synthesis of (*E*)-quinoxalin-2(1*H*)-ones oximes (Figure 21). The reaction consisted of quinoxalin-2(1*H*)-ones, ketone and *tert*-butyl nitrite (TBN) as three components. It should be noted that the reaction cannot be completed in the absence of CH_3_SO_3_ or under a N_2_ atmosphere. A variety of gram-level oxime-containing quinoxalin-2(1*H*)-ones were obtained by recrystallization. The reaction mechanism is proposed by controlled experiments in Figure 21. Initially, the imino ion **18e** was produced by the protonation of **18a**. Simultaneously, acetone **18b** was converted to the enol form **18f** under acidic conditions. Next, A reacted with **18f** to give **18h**. Then, **18h** was turned into **18i** by an oxidative dehydrogenation process. Simultaneously, the homogeneous decomposition of TBN produced *tert*-butoxy (*t*BuO·) and nitroso (·NO) radicals. Subsequently, the nitroso (·NO) radical assaulted the C=C bond of **18i** to give intermediate **18j**, which then oxidized to intermediate **18k** through tert-butoxy (tBuO·). Lastly, the goal product was produced by the isomerization of **18k**.

### 2.5. Simultaneous Construction of Both C–C and C–Cl Bonds via Multi-Component Tandem Reactions

In 2021, Zhang’s group [55] published the second three-component reaction on quinoxalin-2(1*H*)-ones (Figure 22). A novel clean green multicomponent transformation of methyl ketones with α-bifunctionalization was achieved by a homogeneous ion-exchange resin Amberlyst 15 catalyst and sunlight catalysis. The reaction was carried out in water and air. Quinoxalinones containing both chloro- and keto-functional groups were obtained in moderate to acceptable yields. A similar reaction mechanism to that previously reported in Figure 21 was proposed through a series of controlled experiments.

### 2.6. Simultaneous Construction of Both C–C and C–S Bonds via Multi-Component Tandem Reactions

Chemicals and physiologists are very interested in sulfur-containing compounds since they play an instrumental role in drug discovery and structural modification. Sulfones can not only act as wonderful pharmaceuticals, such as anticancers, antioxidants, and antifungals, but also serve as significant precursors for the synthesis of Sulphonic acids, sulfoxides, sulfanilamides and thioethers [56].

#### 2.6.1. Sulfonyl Hydroxylation

In 2019, Koley et al. [57] proposed a three-component radical cascade reaction using TBHP as oxidant for the synthesis of α-sulfonyl-β-heteroaryl scaffolds using olefins, aryl sulfites and quinoxalin-2(1*H*)-ones (Figure 23). When screening the reaction conditions, oxygen was found to be unfavorable for the reaction. A series of quinoxalin-2(1*H*)-ones containing sulfonyl groups were synthesized with high yields and functional group tolerance. A free radical reaction mechanism was postulated by controlled experiments (Figure 23).

In 2020, Lee et al. [58] reported a stereoselective synthesis of various functionalized vinyl sulfones via a three-component vinyl sulfonation reaction of quinoxalin-2(1*H*)-ones, terminal alkynes and sulfonyl hydrazides under copper catalysis and TBHP oxidation. (Figure 24). The reaction had a vast range of substrates. In the absence of a catalyst, this reaction could not proceed. Preliminary studies suggest that the reaction was a single electron transfer (SET) reaction mechanism. Sulfonyl hydrazide can be used as a source of sulfonyl radicals, which form through the interaction between Cu(II) and TBHP (Figure 24).

In 2021, He’s group [59] proposed a visible-light photocatalytic four-component sulfur dioxide insertion reaction using Rhodamine 6G as a catalyst to synthesize a series of vinyl sulfone-containing quinoxalin-2(1*H*)-ones from alkynes, sodium metabisulfite, aryldiazonium and quinoxalin-2(1*H*)-ones (Figure 25). Both C–C bonds and C–S bonds could be constructed in a single operation. By comparing NMR spectra, the vinyl sulfone product was determined to be in the *Z*-configuration, with only trace amounts of isomers in the *E*-configuration (*Z*/*E* > 30/1). A possible mechanism for visible light-induced reactions is proposed in Figure 25. In the same year, their group [60] proposed a similar visible-light photocatalytic four-component sulfur dioxide insertion reaction. The difference is that alkynes are replaced with alkenes.

In 2021, Jiang et al. [56] published a direct bifunctionalized three-component cascade reaction of olefins with quinoxalin-2(1*H*)-one and sodium sulfonate using potassium persulfate (K_2_S_2_O_8_) as an oxidant to obtain a series of sulfone derivatives (Figure 26). This method used only environmentally friendly water as the reaction solvent. A free radical reaction mechanism similar to such reactions was proposed by controlled experiments (Figure 26). Initially, sodium sulfonates were converted to the corresponding radicals (**22e**) by the oxidation of K_2_S_2_O_8_. Subsequently, a radical addition of a sulfone radical via the C=C bond of alkenes was a critical step and led to the formation of the alkyl radical intermediate **22f**. The active alkyl radical species **22f** then reacted with quinoxalinones (**22a**) by addition at the most electrophilic C3 position to give the nitrogen radical species (**22g**), which rapidly underwent further dehydrogenation/oxidation by potassium persulfate to give the bifunctional sulfone derivatives (**22d**).

In 2022, Wei’s group [61] reported the photocatalytic three-component tandem reaction of quinoxalin-2(1*H*)-ones, alkenes, and sulfites (Figure 27). Using 4CzIPN as the catalyst and oxygen (air) as the oxidant, the tandem reaction prepared a series of sulfonated quinoxalin-2(1*H*)-ones. Free radical trapping experiments and fluorescence quenching experiments elucidated the visible light-mediated free radical reaction process (Figure 27).

#### 2.6.2. Thioalkylation

In the same year, Roy et al. [62] first developed an efficient and environmentally friendly three-component cascade reaction for the thioalkylation of quinoxalin-2(1*H*)-ones with olefins and aryl disulfides under visible light in the presence of no catalyst (Figure 28). It is worth noting that oxygen is very detrimental to product generation. A series of sulfur-containing quinoxalin-2(1*H*)-ones were synthesized with excellent functional group tolerance and broad substrate suitability. This approach is also applicable to phenylacetylene instead of olefins. Mechanistic studies suggest that a major pathway is the involvement of quinoxalin-2(1*H*)-ones in the reaction as photosensitizers.

### 2.7. Simultaneous Construction of Both C–C and C–D Bonds via Multi-Component Tandem Reactions

In 2020, Li et al. [63] first reported a fast three-component deuterated quinoxalin-2(1*H*)-ones reaction promoted by Fe(III) with alkenes and NaBD_4_ (Figure 29). A series of deuterated quinoxalin-2(1*H*)-ones were synthesized with moderate to high yields. Through controlled experiments, they proposed a free radical reaction mechanism similar to that of other three-component reactions (Figure 29).

## 3. Summary and Outlook

The direct C–H functionalization of quinoxalin-2(1*H*)-ones via multicomponent reactions allows the one-pot introduction of diverse functional groups, a facile and efficient method that has attracted increasing interest from synthetic chemists and which holds broad promise for medicinal chemistry and functional materials. In this review, we provide an overview of the C–H functionalization of quinoxalin-2(1*H*)-ones at the C3 position via multi-component reactions that have been reported in recent years in terms of the different categories of bond formation, and show the mechanisms of reactions that are mostly radical-based in order to provide a more in-depth and rapid understanding of the reactions. In addition, a number of meaningful functional groups commonly found in drugs such as trifluoromethyl, difluoromethyl, perfluoromethyl, alkyl, cyano, oxime, azide, alkoxy, sulfonyl, sulfur, deuterium and chloride are introduced into these quinoxalin-2(1*H*)-ones are summarized.

Despite these outstanding achievements, relatively few multicomponent reactions of quinoxalin-2(1*H*)-ones have been reported, and further work on the development of multicomponent reactions of quinoxalin-2(1*H*)-ones is essential. Firstly, the multicomponent reactions of quinoxalin-2(1*H*)-ones have not yet involved the simultaneous construction of C–C bonds and C–B/SI/P/F/I/SE bonds. In particular, the C–B/F/I/SE bond at the C3 position of quinoxalin-2(1*H*)-ones, which has not yet been completed, is expected to be completed by the multicomponent reactions of quinoxalin-2(1*H*)-ones. Secondly, the construction of fused ring and macrocyclic compounds by multicomponent reactions of quinoxalin-2(1*H*)-ones is also unexplored. Thirdly, asymmetric synthesis is a constant topic of research in organic methodology and the direct completion of asymmetric catalytic reactions at the C3 position of quinoxalin-2(1*H*)-ones is challenging, whereas the introduction of chiral centers via multicomponent reactions of quinoxalin-2(1*H*)-ones is a concise route. Fourthly, from the perspective of green chemistry, both photocatalysis and non-homogeneous catalysts that can be recycled are indispensable for the development of multi-component reactions of quinoxalin-2(1*H*)-ones. Fifthly, the parent nucleus structure of quinoxalin-2(1*H*)-ones is present in the metabolites of Streptomyces [64] (Figure 2), and it is necessary to investigate the mimetic structure through the multicomponent reactions of quinoxalin-2(1*H*)-ones. In conclusion, we hope that this review will be useful for research in this field. The multicomponent reactions of quinoxalin-2(1*H*)-ones offer an atomically economical, environmentally friendly and sustainable approach for the preparation of a wide range of functionalized quinoxalin-2(1*H*)-ones with promising applications in materials science and medicinal chemistry.

## Data Availability

Not applicable.

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
