# Peer review of "Recent Developments in Direct C–H Functionalization of Quinoxalin-2(1H)-Ones via Multi-Component Tandem Reactions"

_molecules, 2023, doi:10.3390/molecules28062513_

Round 1
Reviewer 1 Report
The recent literature reports numerous reviews regarding the quinoxalin-2-ones and the authors make an effort to differentiate this manuscript, also if a partial overlap is unavoidable. The review is set up graphically with the schemes that precede the text describing the content, but for better reading, the text should precede the corresponding graphical scheme. Moreover, the description regarding the reaction conditions is not always clear. To make the review more readable, I suggest that all the schemes must show: a) the catalyst, b) the oxidant agent, c) the radical source for the radical reaction path. Moreover, the need for a transition-metal catalyst is often overlooked in the describtion (see for example schemes 7, 12, the Ir-catalyst is not cited in the text).
In particular:
- the acronyms used in the text should be explain (as 4CzIPN, TPPT, Acr+MesClO4- etc.)
- scheme 6, intermediate 4h omitted
- scheme 7 and 12 the Ir-catalyst not cited in the text
- scheme 9 the oxidant not cited in the text
- scheme 15, path A and path B needs an explanation
- scheme 17, better explanation of the mechanism is required
- schemes 18 and 19, the same mechanism, the only difference is the presence of base in the second case. A comparison between the 2 results would be appreciable
- scheme 21, compound 18e wrong formula
- scheme 24, the mechanism needs the presence of the oxidant specie
- scheme 25 should be moved next to the scheme 23, due to the same synthetic pathway, starting from alkynes and affording the same product
- schemes 21 and 29, are referred both to the transformation of methyl ketones through the bifunctionalization, the only difference is regarding the reagent to introduce, NO and Cl atom, then the two schemes should stay close together
- Reference 21 must be correct
Some typing mistakes are present in the text
The following references must be added:
Org. Lett. 2019, 325, Adv. Synth. Catal. 2014, 3685; Adv. Synth. Catal. 2018, 4509
Reviewer 2 Report
Direct CH multi-functionalization of quinoxalin-2(1H)-ones is of significant interest for medical chemistry in terms of developing methods for the synthesis of potentially biologically active substances that can be used in medical practice. The chemistry and biological properties of such compounds present researchers with a large number of surprises and amazing discoveries. The review is well structured, the authors discuss the results of the most up-to-date studies published over the past 10 years. The review is written in a good literary language, it is easy to read and with great pleasure. Unfortunately, the reviewer cannot fully assess the quality of the language. Despite the fact that quite a lot has been done in the field under consideration, nevertheless, many new directions have appeared recently, including asymmetric synthesis, green chemistry, etc. I believe that the review may be of interest to a wide range of researchers.
Reviewer 3 Report
The manuscript by Lei and co-workers provides a comprehensive and interesting review concerning the recent development multi-component approaches to direct C–H functionalization of 2-quinoxalin-2(1H)-ones. Overall, this paper is a quit well-written paper but some amendments would be useful to improve its quality. Following are the suggestions:
1) In the "Introduction" section, the authors refer to some reviews concerning the direct C-H functionalization of the C(3) position of quinoxalinone derivatives. However, I would like to note that there is absolutely no mention of the possibility of a direct acid-catalyzed C-H bond functionalization based on the methodology of nucleophilic aromatic substitution of hydrogen (SNH-reactions). I strongly recommend that the authors review this synthetic approach and add the corresponding references: (https://doi.org/10.1016/j.mencom.2017.01.032, https://doi.org/10.15826/chimtech.2022.9.1.03, https://doi.org/10.1070/MC2006v016n01ABEH002153, https://doi.org/10.1007/s11172-016-1319-x, https://doi.org/10.1016/j.ejmech.2021.113577, etc).
2) It would also be worth noting that, in addition to a wide biological application, the quinoxalin-2(1H)-one fragment is a promising electron-withdrawing substituent for the development of push-pull systems with promising photophysical properties for various material science's application (for example, https://doi.org/10.1016/j.dyepig.2019.107580, https://doi.org/10.1016/j.dyepig.2020.108958, etc).
3) The authors should explain of the abbreviations of the compounds shown in Schemes 6 (4СzIPN), 7 and 8 (photocatalysts).
4) The authors use the phrase "thick and macrocyclic compounds" in the text. It remained quite incomprehensible. What do the authors mean by "thick compounds"?
Round 2
Reviewer 1 Report
The revised version of the manuscript satisfies the required modification